# Fusion of Dendritic Cells Activating Rv2299c Protein Enhances the Protective Immunity of Ag85B-ESAT6 Vaccine Candidate against Tuberculosis

**DOI:** 10.3390/pathogens9110865

**Published:** 2020-10-22

**Authors:** Yong Woo Back, Hyun Shik Bae, Han-Gyu Choi, Dang Thi Binh, Yeo-Jin Son, Seunga Choi, Hwa-Jung Kim

**Affiliations:** 1Department of Microbiology, College of Medicine, Chungnam National University, Daejeon 35015, Korea; lenpk@nate.com (Y.W.B.); wdrgasd777@naver.com (H.S.B.); ekdrms20000@gmail.com (H.-G.C.); dangbinh91@gmail.com (D.T.B.); syj1234@cnu.ac.kr (Y.-J.S.); 2Department of Medical Science, College of Medicine, Chungnam National University, Daejeon 35015, Korea

**Keywords:** *Mycobacterium tuberculosis*, multifunctional T cell, T central memory cell, dendritic cell

## Abstract

In *Mycobacterium tuberculosis* infection, naïve T cells that encounter mycobacterial antigens through dendritic cells (DCs) induce various CD4^+^ T-cell responses; therefore, appropriate DC activation is the key for protective immunity against tuberculosis. We previously found that Rv2299c-matured DCs induce Th1 differentiation with bactericidal activity. In this study, to prove that Rv2299c could enhance the protective immunity of other vaccine candidates comprising T-cell-stimulating antigens, Ag85B-ESAT6, a well-known vaccine candidate, was selected as a fusion partner of Rv2299c. Recombinant Rv2299c-Ag85B-ESAT6 protein induced DC maturation and activation. Furthermore, fusion of Rv2299c enhanced the protective efficacy of the Ag85B-ESAT6 vaccine in a mouse model and significantly higher production of TNF-α and IL-2 was detected in the lungs, spleen, and lymph nodes of the group immunized with the Rv2299c-fused protein than with Ag85B-ESAT6. In addition, fusion of Rv2299c enhanced the Ag85B-ESAT6-mediated expansion of multifunctional CD4^+^ T cells. These data suggested that the DC-activating protein Rv2299c may potentiate the protective immunity of the vaccine candidate comprising T cell antigens.

## 1. Introduction

According to the WHO Global Tuberculosis Report 2018, tuberculosis (TB) caused an estimated 1.6 million deaths in 2017. Despite extensive research, TB continues to be a major infectious disease. HIV infection and emergence of multidrug-resistant TB (MDR-TB) are strong challenges to TB control [1]. The Bacillus Calmette–Guérin (BCG) vaccine is mainly used to prevent active TB disease in infants, but it has not been effective in the management of disease in adolescents and adults [1,2]. Consequently, the development of a more effective vaccine is of the highest priority.

*Mycobacterium tuberculosis* (Mtb) infects phagocytic antigen-presenting cells (APCs) in the lung, including alveolar macrophages and dendritic cells (DCs). In these cells, Mtb evades host immunity by inhibiting host defense mechanisms, including phagosome maturation, by secretion of inflammatory cytokines and antigen presentation [3]. In early infection, an understanding of the mechanism of T-cell evasion, including impaired or misregulated dendritic cell maturation and delayed priming of CD4^+^ T cells [4,5,6], is essential in the design of vaccines for successful antigen activation of CD4^+^ T cells. In the lungs, naïve T cells that encounter mycobacterial antigens induce various CD4^+^ T-cell responses through DCs, which bridge innate and adaptive immunity. After lung DCs endocytose antigens and migrate to lymphoid organs, they initiate T-cell differentiation in the lung-draining lymph nodes [7]. The migration of DCs is promoted by interleukin (IL)-12p40 and limited by IL-10 [8,9]. There are many subsets of CD4^+^ T cells whose development is triggered by DCs, such as T-helper 1 (Th1), Th2, Th17, and regulatory T cells (Tregs), which cooperate or interfere with each other to control the balance between active and latent TB [10]. In particular, antigen-specific CD4^+^ Th1 cells play a protective role against TB by producing cytokines such as interferon (IFN)-γ or tumor necrosis factor (TNF)-α, which contribute to the recruitment and activation of innate immune cells such as monocytes [11]. A study of infectious progression in mice lacking IFN-γ showed that Th1-polarized CD4^+^ T cells are more important than Th2-immune response for the control of TB [12,13]. However, although many studies have identified the function of CD4^+^ T cells against TB, the role of T cells in host immunity is not defined. There are several CD4^+^ T cell effector subtypes, from activated T cells that only produce IL-2 or IFN-γ to multifunctional cells expressing IL-2, IFN-γ, and TNF-α [14]. The role of multifunctional T cells is closely related to the regulation of TB infection and protection [9,15,16]. Since the development of these effector T cells initially requires DC activation and migration to the lymph nodes, it is necessary to regulate the activity of DCs to exert protective immunity against TB.

Many studies have focused on the development of multi-subunit vaccines using viral vectors or adjuvants. Although diverse mycobacterial antigens have been investigated for development of TB vaccine, only a few proteins have been selected as vaccine candidates to enter human clinical trials. Among them, antigen 85 complex and ESAT6 are most commonly used [17,18,19]. However, MVA85A did not elicit significant BCG-prime boosting effect in human trial [20], which has led to search a new vaccine target and develop the diverse strategies for vaccine research. Therefore, it is important to search other antigens for replacing T cell-stimulating antigens and enhancing their activity. Several mycobacterial antigens are reported to induce DC maturation and Th1 polarization [21,22,23], but their vaccine potential against TB is yet to explored.

We previously reported a novel DC-activating protein, Rv2299c. Rv2299c-matured DCs induce Th1 differentiation with bactericidal activity, and in particular, the fusion of Rv2299c to ESAT6 enhances the immunoreactivity and BCG prime-boosting efficacy of ESAT6 [24], whereas Rv2299c alone does not confer a significant protective effect. Based on these data, we hypothesized that the DC-activating protein Rv2299c could enhance the protective immunity of other vaccine candidates comprising T-cell-stimulating antigens. In this study, to prove our hypothesis, we selected the fusion protein Ag85B-ESAT6, a well-known TB vaccine candidate, as the fusion partner of Rv2299c. We found that the inclusion of Rv2299c to the Ag85B-ESAT6 fusion protein leads to an increase in the immune reactivities and protective efficacy of Ag85B-ESAT6. In addition, a higher frequency of multifunctional CD4^+^ T cells was found in mice vaccinated with Rv2299c-Ag85B-ESAT6 than in Ag85B-ESAT6. Thus, our results provide a new strategy for the design of vaccine candidates based on DC-activating antigens.

## 2. Results

### 2.1. Rv2299c-Ag85B-ESAT6 Protein Induces DC Maturation

The recombinant Rv2299c-Ag85B-ESAT6 fusion protein (RAE6) was constructed in the pET22b vector (Figure 1a). The fusion protein was produced in *E. coli* BL21 cells and purified under endotoxin-free conditions. After purification and dialysis, the purity of the recombinant protein was confirmed by sodium dodecyl sulfate-polyacrylamide gel electrophoresis and immunoblot assay (Figure 1b). The purified protein had a molecular mass of approximately 120kDa, which is consistent with the molecular weight of the full-length amino acid product. To establish that there was no endotoxin or LPS contamination during protein purification, we examined whether the immunologic effect of the Rv2299c-fused protein was maintained in polymyxin B (PMB) treatment used to minimize endotoxin or LPS contamination. The secretion of MCP-1, a key chemokine of monocytes/macrophages, was not affected in RAW264.7 cells following stimulation with PMB-treated Rv2299c-fused protein (Figure 1c). However, the secretion of MCP-1 was significantly inhibited by stimulation with PMB-treated LPS. This suggests that endotoxin or LPS, which can affect immune responses, was not a contaminant of protein preparation.

We first examined the effect of Rv2299c-fused protein on DC maturation prior to investigating its protective effect. Immature DCs loaded with Rv2299c antigen have been shown to activate DC maturation and Th1 polarization [24]. To determine whether Rv2299c-Ag85B-ESAT6 fusion protein also affects the differentiation of immature DCs, we prepared immature BMDCs that were incubated with GM-CSF and IL-4 under standard conditions for 7 days. We determined pro-inflammatory cytokine secretion during DC maturation by incubation for 24 h in the presence of 10 μg/mL Rv2299c-fused protein or LPS (as a positive control). We found that Rv2299c-fused protein significantly increased secretion levels of pro-inflammatory cytokines, including TNF-α, IL-1β, IL-6, and IL-12p70, by DC maturation (Figure 1d). Differentiation of T cells is determined by antigen-primed DC maturation [25]. Peptides presented by MHC class II molecules on DCs determine the antigen-specificity of the T-cell response via T-cell receptor ligation. Initiation of protective immunity also requires co-stimulation mediated by the triggering of CD28 by CD80 and CD86. Such stimulation generally allows for the differentiation of naïve T cells, which involves high expression of selective cytokines such as IL-12. For optimal expression of T cell-polarizing molecules, the primed DC requires cross-talk with the T cells, involving ligation of CD40 by CD40L after activation by these signals [25,26]. We also investigated the expression of various surface markers of DC maturation using FACS to examine the phenotypic alterations in DCs. Surface markers, including CD80, CD86, CD40, MHC class I, and MHC class II, showed higher expression in DCs treated with Rv2299c-fusion protein than controls (Figure 1e). DCs migrate to the lymph nodes under the influence of IL-12 cytokines and chemokines to drive naïve T-cell differentiation toward Th1 polarization and produce IFN-γ, leading to macrophage activation and bacterial control [11,27]. To analyze whether the polarization and differentiation of T cells is influenced by DCs stimulated by the Rv2299c-fused protein, we examined IFN-γ secretion by DC–T cell co-culture assay. We compared the cytokines production of T cells stimulated by Rv2299c-fused protein-matured DCs and LPS-matured DCs. Splenocytes co-cultured with Rv2299c-fused protein-treated DCs produced significantly greater levels of IFN-γ than DCs matured with LPS (Figure 1f). Syngeneic T cells primed with Rv2299c-fused protein-matured DCs produced high levels of IFN-γ, suggesting that they can drive the Th1 immune response. Based on these data, we found that similar to Rv2299c, the Rv2299c-fused protein can affect the differentiation of immature DCs.

### 2.2. Fusion of Rv2299c Enhances the Protective Efficacy of Ag85B-ESAT6

To determine whether Rv2299c fusion could enhance the protective immunity of Ag85B-ESAT6, we compared their protective effect with an Mtb H37Ra challenge model. It is well known that Mtb H37Rv and H37Ra strains are derived from the same parental strain but differ in their virulence in experimental animals [28]. Owing to the low risk of virulence, we compared the protective immune responses of Rv2299c-Ag85B-ESAT6 protein (RAE6) and Ag85B-ESAT6 protein (AE6) against MtbH37Ra in a mouse model. Before testing, we analyzed the functional activation of DCs using recombinant Ag85B-ESAT6 protein. Pro-inflammatory cytokines and surface markers were significantly upregulated in DCs following treatment with Ag85B-ESAT6 protein and were found to be similar to that after treatment with Rv2299c-Ag85B-ESAT6. However, IL-12p70 production was significantly higher in the cells stimulated with Rv2299c-combined protein (10 ng/mL) than in uncombined Ag85B-ESAT6 (6 ng/mL) (Appendix A). As shown in Figure 2a, 4 weeks after the final immunization, all mice were challenged with Mtb H37Ra via intratracheal administration. To compare protective immunity, we first analyzed the levels of Th1-related cytokines in lung cells from an immunized mouse before Mtb challenge. Lung cells from an immunized mouse were stimulated with Ag85B. We found that production of IFN-γ and IL-2, Th1-related cytokines, was not significantly different in the group immunized with Rv2299c-fused protein upon Ag85B stimulation relative to that after immunization in the uncombined protein group (Figure 2b). Additionally, the levels of TNF-α, a pro-inflammatory cytokine, were also not a significant difference between both groups. In addition, a vaccine-induced Th17 response has been shown to support protection through the recruitment of protective Th1 cells [29,30]. For this reason, we also examined whether IL-17 production increased in the lung cells in the group immunized with the Rv2299c fused protein. However, we could not find a significant difference between the mice immunized with both proteins.

Next, we investigated the protective immune response against Mtb in an immunized mouse group. The mycobacterial burden in the lung was determined at 3 weeks after challenge. The bacterial loads were significantly lower in the groups immunized with both proteins compared to that in the adjuvant control, and significantly lower in the group immunized with Rv2299c-fused protein compared with that of Ag85B-ESAT6 (Figure 2c). CD4^+^ Th1 cells play a functional role in protective immunity against Mtb, producing cytokines such as IFN-γ or TNF-α, which contribute to the recruitment and activation of innate immune cells [11]. Therefore, we also assessed how Th1-related cytokines changed in the lung cells of immunized mice, 3 weeks after Mtb challenge. We found that the production of IFN-γ and IL-17 were not significantly different in the lungs and spleens between the groups immunized with Rv2299c-fused protein or with Ag85B-ESAT6 protein upon Ag85B stimulation (Figure 2d,e). However, significantly higher production of TNF-α and IL-2 was detected in in the lungs and spleens from the group immunized with the Rv2299c-fused protein than with Ag85B-ESAT6 (Figure 2d,e). In particular, for IL-2, we found that the group immunized with Ag85B-ESAT6 showed no significant difference after Mtb challenge, whereas the group immunized with Rv2299c-fused protein showed a two-fold increase, when compared with before challenge. Similar to the effect in the lung cells, in the splenocytes, a significant increase was observed in the TNF-α and IL-2 levels (Figure 2e) after Mtb challenge.

### 2.3. Fusion of Rv2299c Enhances Ag85B-ESAT6-Mediated Expansion of Multifunctional CD4^+^ T Cells and Memory T Cell Response

Many studies on intracellular pathogens have shown that the simultaneous production of IFN-γ, IL-2, and TNF-α by multifunctional CD4^+^ T cells may correlate with protection from infection [14,31]. Therefore, we used FACS to compare the proportion of multifunctional CD4^+^ T cells in immunized mice after Mtb challenge. We performed intracellular staining for IFN-γ, IL-2, and TNF-α in lung cells stimulated with Ag85B. The group immunized with Rv2299c-fused protein showed a ten-fold higher number of multifunctional CD4^+^ T cells than that in the group immunized with Ag85B-ESAT6 protein (from 0.2–0.3% to 1–1.5% of accumulated cytokine-positive cells). In contrast to the control groups, the distribution of double-positive and triple-positive T cells in infection-induced CD4^+^ T-cell responses was higher in the lung cells from the group immunized with the Rv2299c-fused protein than with Ag85B-ESAT6 (Figure 3a).

For subunit protein-assisted vaccines, the resulting T-cell response appears to be dominated by a less-differentiated T central memory cell (T_CM_) response that mainly expresses IL-2 and/or TNF-α [17]. The emergence of multifunctional T cells is highly associated with memory T cells, which is evidence that a vaccine confers memory immunity [32]. To assess whether vaccination with Rv2299c-fused protein can induce T_CM_ cell populations after Mtb challenge, we analyzed the surface expression of CD62L and CD44 on CD4^+^ T cells using flow cytometry [33]. As shown in Figure 3, the expansion of CD4^+^CD44^high^CD62^low^ effector cells was similar between the groups immunized with Rv2299c-Ag85B-ESAT6 and Ag85B-ESAT6. However, the population of CD4^+^CD44^high^CD62^high^ central memory cells was higher in the group immunized with Rv2299c-Ag85B-ESAT6 than in Ag85B-ESAT6 (Figure 3b). These results suggest that Rv2299c-fused protein may induce higher production of multifunctional cytokines in antigen-specific T cells and increase memory response, activating protective immunity against Mtb, than the uncombined protein.

### 2.4. Fusion of Rv2299c Enhances the Protective Immune Response of Ag85B-ESAT6 in Lymph Node

Upon infection, the priming of CD4^+^ T cells in lymphoid organs is delayed [4,5,6]. After Mtb challenge, we examined Th1-related cytokines in lymph nodes using ELISA to further analyze the protective effect of Rv2299c protein. We found no significant difference in the production of IFN-γ in the lymph nodes between the groups immunized with both proteins; similar results were obtained in the lung cells and splenocytes (Figure 2d,e). However, the production of TNF-α and IL-2 was significantly higher in lymph nodes from the group immunized with the Rv2299c-fused protein than with Ag85B-ESAT6 protein (Figure 4a). We found that IL-17 production in the lymph nodes was also higher in the group immunized with the Rv2299c-fused protein than with Ag85B-ESAT6. In addition, the group immunized with Rv2299c-fused protein showed a two-fold higher number of multifunctional CD4^+^ T cells than the group immunized with Ag85B-ESAT6 (from 0.04–0.05 % to 0.1–0.13% of accumulated cytokine-positive cells) (Figure 4b). These results suggest that the protective effect of Rv2299c-fused protein is maintained in the lymph nodes after Mtb challenge.

## 3. Discussion

DCs play an important role in linking the innate and adaptive immune systems, as sensors for microbial invasion. They process and present antigens to activate naïve T lymphocytes. These actions modulate the characteristics of the T-cell response toward either a Th1 or Th2 effector response [34]. For this reason, many mycobacterial antigens have been identified and characterized in DC activation. For example, proteins of proline-glutamate (PE) and proline-proline-glutamate (PPE) families in *M. tuberculosis*, PE_PGRS 17, PE_PGRS 11, and PPE34, interact with TLR2 and trigger the functional maturation of human DCs [35,36]. The cell wall-associated antigen PPE60 (Rv3478) activates DC maturation and T-cell differentiation [37]. In addition, other antigens have also been shown to affect the maturation of DCs [22,38,39]. Recognizing the necessity for DC-activating antigens, we have demonstrated that antigens of *M. tuberculosis* lead to increased expression of costimulatory molecules and pro-inflammatory cytokines via DC maturation and drive the Th1 immune response [40,41,42]. These central roles of mycobacterial antigens in activated DC maturation may provide an attractive strategy for developing highly efficient vaccines.

We constructed a multi-fused protein in which the DC-activating antigen was inserted into a vaccine candidate combining secreted antigens associated with host immunity. One of the constituent antigens, Ag85B, has been shown to be a powerful mycobacterial antigen and a major target of the human T-cell response [43,44]. Ag85B is known to induce partial protection in infected animal models [45,46]. The ESAT6 antigen is the basis for TB diagnosis [47,48] and provides the major target for cell-mediated immunity [49]. Thus, many candidates for subunit vaccines are based on potent T-cell antigens such as Ag85B or ESAT6. Previous studies have indicated that vaccination with a fused protein consisting of Ag85B and ESAT6 (H1) promotes a strong immune response, which has a protective effect against TB in animal infection models [18,50,51]. However, owing to the emergence of drug-resistant TB, particularly MDR-TB and extensively drug-resistant TB, candidates consisting of only T-cell antigens confer limited protection against all forms of TB [52]. In addition, although the risk of TB reactivation is high in patients with latent TB infection, studies are insufficient on therapeutic T-cell antigen vaccines that prevent reactivation. We previously identified the Rv2299c protein, which effectively induces DC maturation and Th1 cell response [24]. In particular, the Rv2299c-ESAT6 fusion protein markedly reduced mycobacterial growth under BCG-boosting conditions. Based on these findings, we hypothesized that the fused protein combining DC-activating antigens may confer increased protective immunity than the uncombined protein. Therefore, we investigated the enhanced protective effect by combining Rv2299c in a fusion protein with Ag85B and ESAT6.

Most studies have focused on the Th1-polarized cytokine IFN-γ as a potential indicator of vaccine efficacy. However, many researchers recognize the need to expand beyond the narrow focus on IFN-γ from clinical trial failures of MVA85A [17,20]. In this study, we showed that vaccination with Rv2299c-Ag85B-ESAT6 led to significantly lower bacterial counts in the lungs after infection than by vaccination with Ag85B-ESAT6 without Rv2299c. However, while we did not find significant differences in IFN-γ secretion from the lung and lymph nodes after infection, significantly higher secretion of TNF-α and IL-2 was detected, after vaccination with the Rv2299c-fused protein than with the uncombined Rv2299c. Consistent with our results, a previous study showed that TNF-α has functional roles in host defense against TB [53]. In the treatment of rheumatic diseases, anti-TNF-α therapy increases susceptibility to TB reactivation [54,55]. In addition, IL-2 is a well-studied cytokine that controls the proliferation and differentiation of inflammatory T cells [56]. This cytokine supports the development of memory T-cell function during infection to aid host defense [57,58]. Our data showed that IL-2 secretion was higher in all tissues including the lungs, lymph nodes, and spleen after vaccination with the Rv2299c-fused protein than with the uncombined protein. IL-2 plays an important role in the survival of CD4^+^ memory T cells [59] and is, therefore, closely related to T_CM_ [17]. In protective immunity against TB, CD4^+^ T cells differentiate into T_CM_ cells that migrate to secondary lymphoid organs in the lung, which is likely to be related to long-term protection [17,60]. After vaccination with Rv2299c-fused protein, we observed an increase in the CD4^+^ T-cell population expressing the T_CM_ marker (CD62L^+^ CD44^+^).

CD4^+^ T cell-mediated immunity is associated with the control of mycobacterial infection. Recent studies have shown that the effect of a vaccine that triggers a protective CD4^+^ T-cell response for successful suppression of TB is associated with the induction of multifunctional CD4^+^ T cells expressing IFN-γ, IL-2, and TNF-α [14,61,62,63]. The lungs from mice vaccinated with Rv2299c-Ag85B-ESAT6 showed a higher number of multifunctional CD4^+^ T cells than that in the lungs of mice vaccinated with Ag85B-ESAT6. After infection, the accumulation of activated lung CD4^+^ T cells is delayed by Mtb-mediated inhibition of early antigen presentation in the lymph nodes [5,64,65]. However, a high number of multifunctional CD4^+^ T cells was found in the lymph nodes obtained from mice vaccinated with Rv2299c-fused protein, similar to that in the lung tissues. Therefore, we speculated that Rv2299c, a DC-activating antigen, may potentiate the protective immunity of T cell-antigen to expand multifunctional cytokine producing T cells, conferring protective immunity against TB. Nonetheless, this study had some limitations. There was no test of the BCG-boosting effect or its efficacy against high-virulence mycobacterial strains, each of which should be the focus of future experiments.

## 4. Materials and Methods

### 4.1. Mice

Specific-pathogen-free (SPF) female C57BL/6 mice, aged 6–8 weeks were purchased from Nara Biotech (Seoul, Korea) and used in our study. All animals were housed at the preclinical Research Center of Chungnam National University Hospital (Daejeon, Korea) and fed on sterile food and water ad libitum. The experiments were performed in accordance with the approval of the Institutional Research and Ethics Committee at Chungnam National University (approval number: 201903A-CNU-5) and the guidelines of the Korean Food and Drug Administration.

### 4.2. Preparation of Recombinant Protein

To produce recombinant Rv2299c-Ag85B-ESAT6 fusion protein, the corresponding gene was amplified using PCR using Mtb H37Rv (ATCC 27294) genomic DNA as a template. The primer sequences used were as follows: Rv2299c-5′NdeI-CATATGAACGCCCATGTCGAGCAGTTG; Rv2299c-3′EcoRI-GAATTCGGCAAGGTACGCGCGAGACGTTC; Ag85B-5′ EcoRI-GTACCTTGCCGAATTCGATGTTCTCCCGGCCGGGGC; Ag85B-3′HindIII-GCTCTGTCATAAGCTTGCCGGCGCCTAACGAACT; ESAT6-5′ HindIII-AAGCTTATGACAGAGCAGCAGTGGAAT; and ESAT6-3′XhoI-CTCGAGTGCGAACATCCCAGTGACGTT.

The Rv2299c-Ag85B-ESAT6 gene was inserted into a pET22b (+) plasmid (Novagen, Madison, WI, USA). The plasmid construct was transformed into *Eschrichia coli* BL21 cells and the resultant products were produced and purified as described previously [24]. The recombinant protein was purified using Ni–NTA resin (Qiagen, Valencia, CA, USA). To remove bacterial endotoxins, the dialyzed recombinant protein was incubated with polymyxin B (PMB)-agarose (Sigma) for 6 h at 4 °C and the purified endotoxin-free recombinant protein was filter-sterilized and frozen at −70 °C. The protein concentration was estimated using a bicinchoninic acid protein assay kit (Pierce, Rockford, IL, USA).

### 4.3. Cell Culture

BMDCs were generated from C57BL/6 mice aged 5–6 weeks according to a previously described procedure [24]. Briefly, BM cells were differentiated for 7 days in RPMI medium supplemented with 10% fetal bovine serum (FBS), 100 U/mL penicillin/streptomycin, 0.1 mM nonessential amino acids, 50 μM β-mercaptoethanol, 1% HEPES, 1 mM sodium pyruvate, 20 ng/mL granulocyte-macrophage colony-stimulating factor (GM-CSF), and 10 ng/mL IL-4. Flow cytometric analysis revealed that >90% of cells were CD11c+ after differentiation. RAW264.7 cells were maintained in Dulbecco’s modified Eagle’s medium supplemented with 10% FBS, 1% HEPES, and 1% L-glutamine at 37 °C with 5% CO2.

### 4.4. Confirmation of Lipopolysaccharide (LPS) Decontamination of Proteins

Pretreatment with PMB (Sigma) was performed to confirm that the activation of immune cells induced by purified protein was not owing to LPS contamination. RAW264.7 cells (5 × 10^4^ cells/wells) were preincubated with 50 μg/mL PMB for 1 h at room temperature prior to treatment with 100 ng/mL LPS and 10 μg/mL Rv2299c-Ag85B-ESAT6. After 24 h, the monocyte chemoattractant protein (MCP)-1 levels in the supernatant of RAW264.7 cells were analyzed using enzyme-linked immunosorbent assay (ELISA).

### 4.5. Cytokine Assays

A sandwich ELISA kit (eBioscience, San Diego, CA, USA) was used to detect inflammatory cytokine levels in culture supernatants as previously described [24].

### 4.6. In Vitro T-Cell Response Assay

BMDCs (2 × 10^6^ cells) were stimulated with LPS or Rv2299c-Ag85B-ESAT6 for 24 h. Then, each stimulated BMDCs (1 × 10^5^ cells) were co-cultured with splenocytes (1 × 10^6^ cells) at a ratio of 1:10. After 3 d, supernatants were harvested and cytokine production was analyzed using ELISA.

### 4.7. Flow Cytometry

To investigate the surface molecules in DCs after 24 h of LPS or Rv2299c-Ag85B-ESAT6 protein treatment, the CD11c-positive cells were stained with phycoerythrin-conjugated anti-CD80, anti-CD86, anti-H-2Kb [18], and anti-I-Ab (MHC class II) from eBioscience. Fluorescence was measured by fluorescence-activated cell sorting (FACS) using flow cytometry.

To create single-cell suspensions, harvested lung cells and cervical lymph node tissue were incubated in RPMI digestion media according to a previously described procedure [66]. Single-cell suspensions of the lungs and cervical lymph nodes of immunized mice were stimulated with Ag85B (2 μg/mL) for 24 h or 12 h at 37 °C in the presence of GolgiStop (BD Biosciences) according to a previously described procedure [66]. The cells were first blocked with Fc Block (eBioscience) for 15 min at 4 °C and then stained with fluorophore-conjugated antibodies purchased from eBioscience. Cells stained with appropriate isotype-matched immunoglobulins were used as negative controls. The cells were fixed and permeabilized using a Cytofix/Cytoperm (BD Biosciences) according to the manufacturer’s instructions. The cells were stained in a permeation buffer with fluorochrome-conjugated flow cytometry antibodies purchased from eBioscience. Then, the samples were detected on the NovoCyte with FACSDiva and analyzed using FlowJo software (Tree Star, Ashland, OR, USA).

### 4.8. Immunization Procedure

Female C57BL/6 mice, aged 6 weeks, were subcutaneously injected thrice with 0.2 mL of Rv2299c-Ag85B-ESAT6 (5 μg) or Ag85B-ESAT6 (5 μg) at 2-week intervals using dimethyldioctadecylammonium liposomes (DDA/250 μg; Sigma) containing monophosphoryl lipid-A (MPL/25 μg; Sigma). The protein was a mixture with DDA and MPL according to a previously described procedure [19]. Briefly, the Protein mixture with DDA was heated in a 70 °C for 30 s, then sonicated for 30 sec. This step was repeated twice. Before use finally, MPL was mixed.

### 4.9. M. tuberculosis Infection in Mice

To verify the vaccine effect, mice (after 4 weeks of the last vaccination) were infected with MtbH37Ra. Briefly, following anesthetization with 1.2% 2,2,2-tribromoethanol (Avertin; Sigma), exposed via a small midline incision, and each mouse was inoculated intratracheally with 1 × 10^6^ CFU/50 μL of Mtb H37Ra. TB infections by the intratracheal route were performed using the Biological Safety Cabinet Class II B2 type (ESCO, Seoul, Korea). To analyze the immune response, mice were euthanized 4 weeks after the last vaccination or 3 weeks after infection.

### 4.10. Immunoassay and Bacterial Counts in Mice

The cells from each lung and cervical lymph nodes were stimulated with Ag85B (2 μg/mL) for 12 or 24 h. Subsequently, cells and supernatants were harvested, and flow cytometry and ELISA were performed. The number of bacteria in the spleen or lung was determined using serial 3-fold dilutions of individual whole-organ homogenates, in duplicate, in 7H10 medium. Colonies were counted after 3 weeks of incubation at 37 °C. Protective efficacies were expressed as log_10_ bacterial counts in immunized mice and were compared to the bacterial counts in the infection controls.

### 4.11. Statistical Analyses

All experiments were performed using GraphPad Prism 5 software (GraphPad Software, San Diego, CA, USA). The significance of the differences between samples was evaluated using one-way ANOVA followed by Tukey’s multiple comparison analysis.

## 5. Conclusions

Subunit vaccines such as H4/IC31, H56/IC31 [67], ID93/GLA-SE and M72/AS01E are currently in clinical trials. Some of these vaccines are also aimed to prevent TB reactivation in LTBI patients as therapeutic vaccines. ID93 combining Rv1813, Rv2608, Rv3619 and Rv3620 successfully completed Phase 2a trials by binding the TLR-4 agonist adjuvant GLA-SE. [20]. The final analysis of phase IIb clinical trial after exposure of M72/ASO1E showed high efficacy against tuberculosis progression in patients infected with *M. tuberculosis* [21]. ESAT6-based subunit vaccines have the potential to confuse a new generation of diagnostic tests that are routinely used to diagnose potential *M. tuberculosis* infection [22]. Previous papers reported that *M. tuberculosis* impairs the antigen-presenting function of DCs [5,6]. Although finding new antigens which can effectively activate DC is in great demand, it is not sufficiently reflected in the aforementioned vaccine candidates. In summary, our findings highlight the importance of vaccine construction using DC-activating antigens. While it remains to be seen whether a BCG-boosting protective immune response can be elicited against a high-virulence strain by DC-activating combined-antigen vaccine candidates, we demonstrate a new perspective on the design of fused proteins as an attractive strategy for developing highly efficient vaccines. Therefore, we are currently searching the promising vaccine candidates as a fusion partner of Rv2299c, and planning to validate their BCG-prime boosting efficacy.

## Figures and Tables

**Figure 1 pathogens-09-00865-f001:**
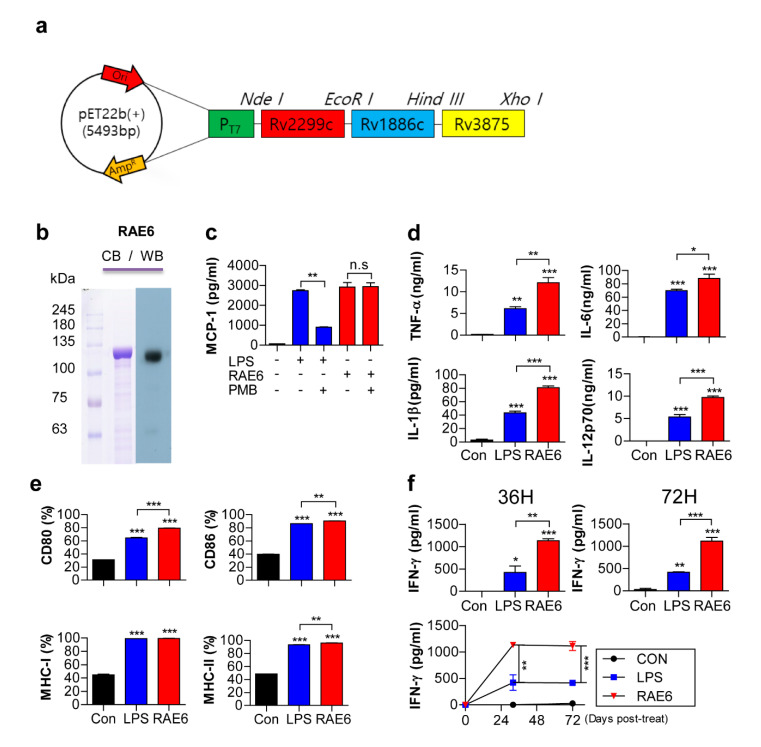
Rv2299c-Ag85B-ESAT6 induces maturation of dendritic cells (DCs) and Th1 responses. (**a**) Schematic diagram of Rv2299c-Ag85B-ESAT6 fusion protein (RAE6). (**b**) Recombinant Rv2299c-Ag85B-ESAT6 was produced in BL21 cells and purified using Ni-NTA resin. The purified protein was subjected to sodium dodecyl sulfate-polyacrylamide gel electrophoresis (SDS-PAGE) and western blot (WB) analysis using a mouse anti-His antibody (Ab). (**c**) To ensure that Rv2299c-Ag85B-ESAT6-induced RAW264.7 cells (5 × 10^4^ cells) activation was not owing to endotoxin contamination in the protein preparation, Rv2299c-Ag85B-ESAT6 (10 μg/mL) or lipopolysaccharide (LPS, 100 ng/mL) was pretreated with polymyxin B (20 μg/mL) for 1 h prior to stimulation of RAW264.7 cell cultures. After 24 h, the quantities of MCP-1 in the culture medium were measured by enzyme-linked immunosorbent (ELISA) assay. All data are expressed as the mean values ± standard deviation (SD; *n* = 3); ** *p* < 0.01 = a significant difference compared to the Rv2299c-Ag85B-ESAT6-treated RAW264.7 cells, as determined by unpaired Student’s t-test. Treatments with no significant effect are indicated as n.s. (**d**) Analysis of functional activation of DCs by Rv2299c-Ag85B-ESAT6. Immature DCs (5 × 10^5^ cells) were cultured in the presence of 10 μg/mL Rv2299c-Ag85B-ESAT6 or 100 ng/mL LPS for 24 h. The quantities of TNF-α, IL-1β, IL-6, and IL-12p70 in the culture supernatant were determined by ELISA. All data are expressed as mean ± SD (*n* = 3). The levels of significance (* *p* < 0.05, ** *p* < 0.01, or *** *p* < 0.001 determined by one-way analysis of variance [17]) of the differences between the treatment data and the control data are indicated. (**e**) The cells were gated on CD11c. The DCs were stained with anti-CD80, anti-CD86, anti-MHC class I, or anti-MHC class II monoclonal Abs (mAbs). The percentage of positive cells is shown in each panel. The bar graphs depict data as the mean ± SD (*n* = 3). The levels of significance (** *p* < 0.01 or *** *p* < 0.001, determined by one-way ANOVA) of the differences between the treatment data and the control data are indicated. (**f**) To examine Th1 differentiation by Rv2299c-Ag85B-ESAT6-treated DCs, naïve splenocytes activated by unstimulated DCs, LPS-stimulated DCs, or Rv2299c-Ag85B-ESAT6-stimulated DCs were co-cultured for 3 days with splenocytes of naïve mice, at a DC to T cell ratio of 1:10. The quantities of IFN-γ in the culture supernatant were determined by ELISA. The data shown are the mean values ± SD (*n* = 3); * *p* < 0.05, ** *p* < 0.01, and *** *p* < 0.001 compared to each group.

**Figure 2 pathogens-09-00865-f002:**
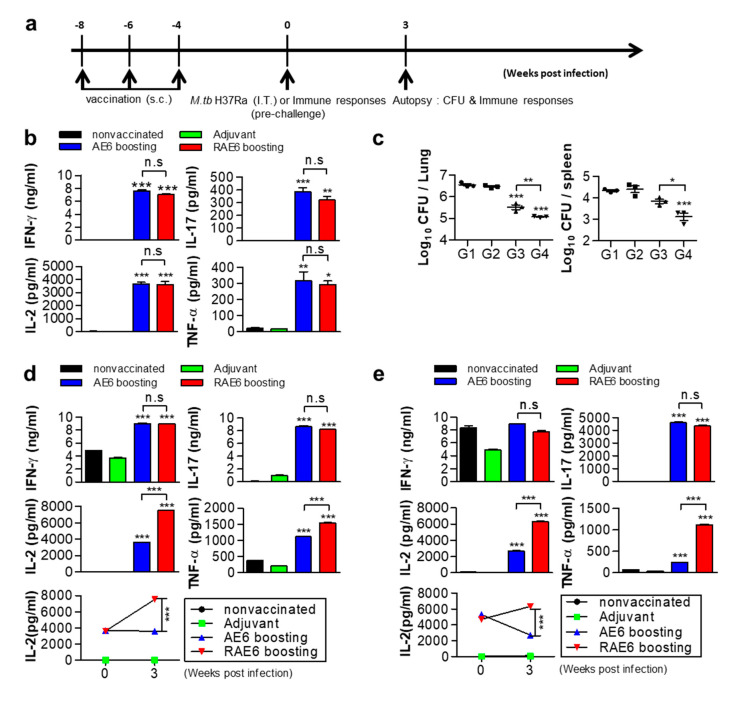
Rv2299c-Ag85B-ESAT6 induces Ag85B-specific Th1 responses and protects the lungs of mice. (**a**) Schematic diagram of the experimental design. (**b**) Lungs after 4 weeks of the last vaccination or 3 weeks post-challenge. (**c**) Three weeks post-challenge, the mice were sacrificed and the bacterial burden (CFU) was measured in the lung. (**d**) lungs and (**e**) spleen were harvested from mice in each group. Each group of cells (10^6^ cells) was treated with Ag85B (2 μg/well) for 24 h. IFN-γ, TNF-α, IL-2, and IL-17 levels were analyzed by enzyme-linked immunosorbent assay (ELISA). The data are shown as the mean ± standard deviation (SD) of 3 samples. One representative plot from three independent experiments is shown. * *p* < 0.05, ** *p* < 0.01, and *** *p* < 0.001, compared to the non-vaccinated control or Ag85B-ESAT6 group. Treatments with no significant effect are indicated as n.s. Group G1: non-vaccinated control; G2: adjuvant alone; G3: Ag85B-ESAT6 (AE6); and G4: Rv2299c-Ag85B-ESAT6 (RAE6).

**Figure 3 pathogens-09-00865-f003:**
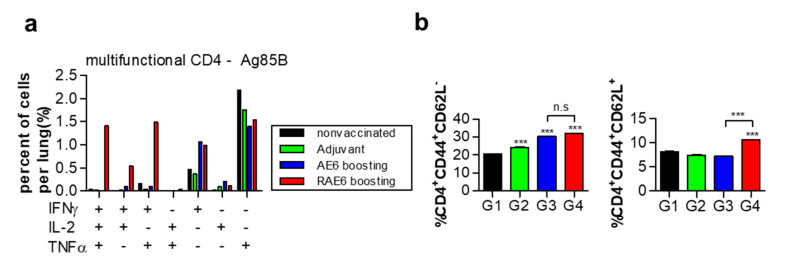
Rv2299c-Ag85B-ESAT6 induces Ag85B-specific Th1 responses and expansion of the effector/memory T-cell population in the lungs of mice. (**a**) Three weeks post-challenge, lung cells (10^6^ cells) were stimulated with Ag85B (2 μg/well) for 12 h at 37 °C in the presence of GolgiStop. The percentage of antigen-specific CD4^+^ T cells producing IFN-γ, TNF-α, and/or IL-2 among the cells isolated from the lungs of each group of mice were analyzed by multicolor flow cytometry by gating for CD4^+^ T cells. (**b**) Lung cells (10^6^ cells) were treated with Ag85B (2 μg/well) for 24 h. The lung cells were stained with anti-CD4, anti-CD62L, and anti-CD44 monoclonal antibodies (mAbs). Bar graphs show antigen-specific CD4^+^CD62L^–^CD44^+^ T-cell or CD4^+^CD62L^+^CD44^+^ T-cell populations among the lung cells. The data are shown as the mean ± SD of three samples. One representative plot from three independent experiments is shown. *** *p* < 0.001, compared to the non-vaccinated control or Ag85B-ESAT6 group. Group G1: non-vaccinated control; G2: adjuvant alone; G3: Ag85B-ESAT6 (AE6); and G4: Rv2299c-Ag85B-ESAT6 (RAE6).

**Figure 4 pathogens-09-00865-f004:**
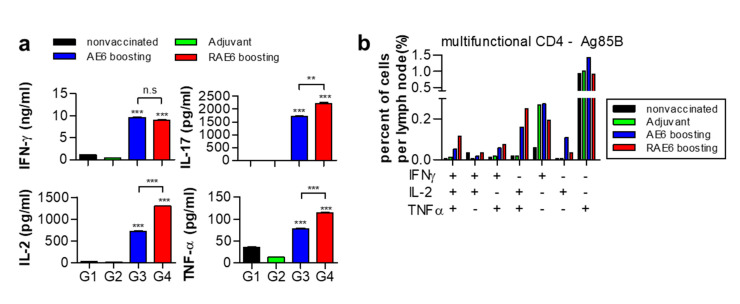
Rv2299c-Ag85B-ESAT6 induces Ag85B-specific Th1 responses in lymph nodes. (**a**) Three weeks post-challenge, cervical lymph nodes were harvested from mice in each group. The lymphocytes (10^6^ cells) were treated with Ag85B (2 μg/well) for 24 h. IFN-γ, TNF-α, IL-2, and IL-17 cytokine levels were analyzed by ELISA. The data are shown as the mean ± standard deviation (SD) of 3 samples. One representative plot from three independent experiments is shown. ** *p* < 0.01 and *** *p* < 0.001, compared to the non-vaccinated control or Ag85B-ESAT6 group. Treatments with no significant effect are indicated as n.s. (**b**) Three weeks post-challenge, lymphocytes (10^6^ cells) were stimulated with Ag85B (2 μg/well) for 12 h at 37 °C in the presence of GolgiStop. The percentage of antigen-specific CD4^+^ T cells producing IFN-γ, TNF-α, and/or IL-2 among the cells isolated from the lymph nodes of each group of mice were analyzed by multicolor flow cytometry by gating for CD4^+^ T cells.

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
