# Peer review of "Fusion of Dendritic Cells Activating Rv2299c Protein Enhances the Protective Immunity of Ag85B-ESAT6 Vaccine Candidate against Tuberculosis"

_pathogens, 2020, doi:10.3390/pathogens9110865_

Round 1

Reviewer 1 Report

  • It would be interesting to admix Rv2299 with Ag85B-ESAT6 to see if the immunogenicity of the antigen Ag85B-ESAT6 will be increased rather than being covalently fused to the antigen. If it worked it would simplify the manufacturing process and also it would also make it possible to change the ratio of Rv2299 to the antigen in the vaccine preparation for an optimised immune response.
  • On the same note, it would also be interesting to admix Rv2299c protein to the BCG vaccine to see if it could increase the immunogenicity and efficacy of the BCG vaccine.
  • The authors could probably determine if Rv2299c induce antibody against itself and if the presence of anti-Rv2299 antibodies has any impact on the immune cells and animals.
  • Figure 1. Consider that the production of INF-gamma, IL-2 and Th-1-related cytokines was not significant different between RAE6 and AE6 before Mtb challenge (see Figure 2), it would be interesting to investigate if AE6 in addition to RAE6 matures DC.
  • Figure 3. “alpha” is missing next to TNF in the X-axis title.
  • Line 79 on Page 2, “PMB” should be spelt out when it is used for the first time.
  • Line 160, what does “not slightly higher” mean?

Author Response

Thank you very much for your letter and for the reviewers’ comments concerning our manuscript. Please see the attachment.

Reviewer 2 Report

In this paper, the authors evaluated the protective efficacy of a protein Rv2299c–Ag85B–ESAT6 against Mtb H37Ra strain obtained by the fusion of dendritic cells activating protein Rv2299c with a TB vaccine candidate, Ag85B-ESAT6. The in vivo studies on mice showed that treatment with fused protein resulted in higher production of TNF-alpha and IL-2 cytokines and improved expansion of multifunctional CD4+ T cells as compared to Ag85B–ESAT6. Although the fused protein did not increase other cytokines production, these studies will help design better vaccine candidates for targeting Mtb strains.

Minor comments:

Please improve the introduction section.

Fig. 2. caption, line 2, change (c) to (b)

Author Response

(The authors gave the same response as above.)

Reviewer 3 Report

In this manuscript, the authors fused Rv2299c with Ag85B–ESAT6, to produce a recombinant protein. The recombinant protein induced DC maturation and activation and, when used as a vaccine in a mouse model, induced significantly higher production of TNF-alpha and IL-2 in the lungs, spleen, and lymph nodes. The authors observed the expansion of multifunctional CD4+ T cells and suggested that the recombinant protein may potentiate the protective immunity of T cells, including the expansion of multifunctional cytokine producing T cells, which could confer protective immunity against TB. The study was well conducted, with the appropriate controls in place. I have no major concerns about this manuscript. However, there are two points I would like to make:

1) The ESAT6, a member of the Region of Difference-1 (RD1), is absent from Mycobacterium bovis bacillus Calmette-Guerin (BCG). Therefore, a vaccine containing ESAT6 it will not be able to boost BCG. The authors did not evaluate the BCG-boosting effect of this vaccine and propose that this should be done in future experiments. Even though DC-activating combined-antigen vaccine candidates could improve the efficacy of BCG, these antigens should be carefully selected. Therefore, the authors should include in the Discussion or Conclusions that DC-activating combined-antigen vaccine candidates could improve the efficacy of BCG with appropriate antigens, but excluding RD1 antigens such as ESAT6.

2) The Conclusions are very short. Given the recent successes in experiment vaccines against TB (M72/AS01E; intravenous BCG in non-human primates; BCG revaccination), the authors should also refer to the advantage of this vaccine compared to the vaccines currently under evaluation.

Minor comments:

3) The manuscript does not show clearly the recombinant protein is LPS-free, because there is no quantification using the gold standard LAL test for LPS detection. Line 83, please replace “This showed that endotoxin” with “This suggests that endotoxin”. Alternatively, the authors can perform the LAL test in the recombinant protein and include the results on the manuscript.

4) In the manuscript, the authors mention the usage of an adjuvant in combination with the recombinant protein. Could the authors please include information in the manuscript about the origin of the adjuvant, how it was produced and who supplied it, including references?

5) In the figure 2c, please show the individual values of CFU for each mouse (lung and spleen). As an example, please look at Figs 3B, C in PMID: 29661928 or Fig 4B, C, D, E in PMID: 31908851.

6) In the methods section, the M. tuberculosis infection in mice, could the authors please clearly refer which type of biosafety facilities were used for these procedures?

Author Response

(The authors gave the same response as above.)
